# Callus Type, Growth Regulators, and Phytagel on Indirect Somatic Embryogenesis of Coffee (*Coffea arabica* L. var. Colombia)

**DOI:** 10.3390/plants12203570

**Published:** 2023-10-14

**Authors:** Consuelo Margarita Avila-Victor, Enrique de Jesús Arjona-Suárez, Leobardo Iracheta-Donjuan, Jorge Manuel Valdez-Carrasco, Fernando Carlos Gómez-Merino, Alejandrina Robledo-Paz

**Affiliations:** 1Colegio de Postgraduados, Campus Montecillo, Carretera México-Texcoco Km. 36.5, Montecillo, Texcoco C.P. 56264, Estado de México, Mexico; consueavic@gmail.com (C.M.A.-V.); arjona@colpos.mx (E.d.J.A.-S.); jvaldez@colpos.mx (J.M.V.-C.); fernandg@colpos.mx (F.C.G.-M.); 2Instituto Nacional de Investigaciones Forestales, Agrícolas y Pecuarias, Campo Experimental Rosario Izapa, Carretera Tapachula-Cacahoatán Km. 18, Tuxtla Chico C.P. 30870, Chiapas, Mexico; iracheta.leobardo@inifap.gob.mx

**Keywords:** callus, somatic embryos, coffee, in vitro culture

## Abstract

Coffee is a crop of global relevance. Indirect somatic embryogenesis has allowed plants of different coffee genotypes to be massively regenerated. The culture medium composition can affect the calli characteristics that are generated and their ability to form somatic embryos. This research aimed to determine the influence of the type of callus, growth regulators, and phytagel concentration on the embryogenic capacity of the Colombia variety. Leaf explants were cultured on Murashige and Skoog medium with 2,4-dichlorophenoxyacetic acid (2,4-D) (0.5–1.0 mg L^−1^), benzylaminopurine (BAP, 1.0 mg L^−1^), and phytagel (2.3–5.0 g L^−1^). The explants generated two types of calli: friable (beige, soft, watery, easy disintegration, polyhedral parenchyma cells) and compact (white, hard, low water content, difficult disintegration, elongated parenchyma cells). About 68% of the total callus generated was compact; this type of callus produced a greater number of embryos (71.3) than the friable one (29.2). The number of differentiated embryos was significantly affected by the concentration of phytagel; higher concentrations (5.0 g L^−1^) resulted in larger quantities (73.7). The highest number of embryos (127.47) was obtained by combining 1.0 mg L^−1^ 2,4-D, 1.0 mg L^−1^ BAP, 5.0 g L^−1^ phytagel, and compact callus.

## 1. Introduction

Coffee is one of the most important crops worldwide due to its commercial and agricultural value. Even though the genus *Coffea* comprises several species, *C. arabica* and *C. canephora* var. Robusta are the most cultivated and traded in the world [1,2,3].

The genus *Coffea* presents limitations for its genetic improvement through conventional programs due to its perennial nature and differences in its level of ploidy and incompatibility; it can take at least 20 years to have a new genotype on the market. In most developing countries, the production of improved and/or certified coffee plants does not exist, which can cause a reduction in productivity and quality [4]. The factors that have most affected coffee production in countries like Mexico are the age of the plantations and their lack of renewal, as well as the incidence of diseases caused by fungi such as *Hemileia vastatrix* Berk & Broome (coffee rust) [5].

It is estimated that the plant requirements to renew coffee plantations in Latin America add up to several hundred million, a demand that could hardly be covered with conventional coffee propagation methods such as cuttings, grafts, and seeds, which so far are inefficient [4]. Renewing plantations with healthy materials, preferably disease-resistant, will contribute to improving coffee production. Colombia is a coffee variety that is resistant to coffee rust. Its plants are small in size and show phenotypic homogeneity; it maintains productivity in both intensive and extensive cultivation and has good grain attributes and drink quality [6].

Somatic embryogenesis is a biotechnological tool that has been used for the large-scale regeneration of coffee plants [7,8,9,10,11,12]. This technique consists of the production of embryos from somatic cells without fertilization. Through this process, the somatic cells of the explant undergo a change in their development program and acquire the capacity to form embryos [13]. There are two ways to develop somatic embryos: direct and indirect routes. The indirect route is the most used method in the protocols developed for different coffee species due to the high multiplication rates that can be obtained. During indirect somatic embryogenesis, cells that have the ability to form embryos develop from a primary callus [9,14,15].

Since the 1990s, somatic embryogenesis has allowed the propagation of commercially important varieties and hybrids, both *Coffea arabica* and *Coffea canephora*. Millions of plants regenerated by this technique have been grown on plantations in Latin America, Africa, and Asia, mainly. In Latin America, where 80% of the world’s coffee production is concentrated, several hundred thousand coffee plants have been produced by somatic embryogenesis. From 2008 to 2018, in Nicaragua, the ECOM trade group, in association with The French Agricultural Research Center for International Development (CIRAD), produced more than 7 million somatic embryos and exported several hundred thousand hybrid plants to countries in Central and South America. In Mexico, Nestlé and the Mexican Agricultural Research Institute (INIFAP) transferred the technology to produce somatic embryos of different varieties of *C. canephora* to the company Nature Source Improved Plants (NSIPs) [4].

On the other hand, Nic-Can et al. [16] point out that not all cells have the capacity to change their cell fate and give rise to somatic embryos. This capacity may depend on the genotype, origin of the explant, composition of the culture medium (salts, growth regulators, gelling agent), physiological state of the donor plant, and the physical conditions to which the cultures are exposed [17].

Growth regulators such as auxins and cytokinins are considered key components of the culture medium to induce somatic embryogenesis [18]. 2,4-Dichlorophenoxyacetic acid (2,4-D) is an auxin analogue that, alone or in combination with cytokinins, has been used to induce somatic embryogenesis in numerous species, including coffee [7,10,19,20,21,22]. Likewise, the cytokinins most commonly used for inducing somatic embryogenesis in coffee are 6-benzyladenine (BAP), 6-furfurylaminopurine (kinetin), zeatin, or N^6^-2-isopentenyl adenine (2iP) [10,19,23].

The requirements for growth regulators to induce somatic embryogenesis depend mainly on the genotype and the physiological state of the explant [10,17]. In different varieties of *C. arabica* and *C. canephora,* it has been possible to regenerate somatic embryos by supplementing the culture media with cytokinins (2iP, BAP) in a greater proportion than auxins (2,4-D, IBA) or with the same concentration of both growth regulators [10,19,20,22,24].

Nonetheless, some authors indicate that the type and concentration of the gelling agent may be as important as the medium’s nutrient salts and growth regulators to induce somatic embryogenesis. Phytagel functions as a gelling agent in the culture medium. It not only provides support to the explant but also affects the water potential of the culture medium. Phytagel allows the explant to absorb water and nutrients to a greater or lesser extent, depending on its concentration. It is known that high concentrations of phytagel could cause water stress in cells. Several authors consider that stress is one of the factors that can promote the reprogramming of cells so that they acquire embryogenic competence [17,25,26,27].

Similarly, the characteristics of the regenerated calli could affect their ability to develop somatic embryos [9,28]. Some authors classify the calli obtained in coffee as embryogenic, non-embryogenic, friable, or compact [12,29]. Knowing the morphological and/or anatomical characteristics of the primary callus generated during the somatic embryogenesis induction stage can help to distinguish those cells with embryogenic capacity from the early stages of culture. The above, in addition to saving time and material, will contribute to improving protocols and reducing costs [30].

Therefore, this research aimed to determine the effect of callus type, culture medium (auxin and cytokinin concentration), and phytagel concentration on the embryogenic capacity of *C. arabica* var. Colombia.

## 2. Results

### 2.1. Effect of Culture Medium and Phytagel Concentration on Embryogenic Callus Formation

The explants cultivated in the induction medium generated two types of calli, which showed different morphological characteristics: friable (beige in color, soft in texture, watery, and easy to disintegrate) or compact (white in color, hard in texture, low in water content, and difficult to disintegrate). Even though no significant differences were found in the amount of friable or compact callus formed in the different combinations of medium and phytagel, it was possible to observe that the predominant type of callus generated was compact. The explants cultivated in the combination of medium M1 with 5.0 g L^−1^ phytagel and medium M2 with 2.3 g L^−1^ phytagel produced 75% of compact callus (Figure 1). Thus, of the total calli obtained, 31.56% were friable and 68.44% were compact.

### 2.2. Formation of Somatic Embryos

#### 2.2.1. Effect of Phytagel Concentration

The number of embryos formed per gram of callus was significantly affected by the concentration of phytagel in the culture medium. When increasing the concentration of phytagel from 2.3 to 5.0 g L^−1^, the number of embryos obtained per gram of callus has doubled (Figure 2a).

#### 2.2.2. Effect of Culture Medium

The number of embryos formed from the calli obtained in the M2 medium (1.0 mg L^−1^ 2,4-D; 1.0 mg L^−1^ BAP) was higher (67.75) than that observed in the M1 medium (0.5 mg L^−1^ 2,4-D; 1.0 mg L^−1^ BAP), without significant differences between the two (Figure 2b).

#### 2.2.3. Effect of Callus Type

The compact calli, like the friable calli, could form embryos; compact calli produced 2.4 times more embryos than friable ones (Figure 2c).

#### 2.2.4. Effect of Callus Type and Phytagel Interaction

It was possible to find significant differences regarding the number of embryos formed per gram of callus as a result of the interaction between the phytagel concentration and the type of callus. The highest number of somatic embryos (112.05) was obtained when the compact callus was cultivated in the presence of 5.0 g L^−1^ phytagel, and the lowest number was obtained when cultivating the friable callus with 5.0 g L^−1^ phytagel (28.23) (Figure 3a).

#### 2.2.5. Effect of the Interaction of Culture Medium and Callus Type

The number of embryos obtained by the interaction between the M2 medium (1.0 mg L^−1^ 2,4-D and 1.0 mg L^−1^ BAP) and the compact callus (85.81) was not statistically different from that obtained by combining the M1 medium (0.5 mg L^−1^ 2,4-D and 1.0 mg L^−1^ BAP) and compact callus and the M2 medium with friable callus, but it was significantly higher than that observed with the interaction between the M1 medium and the friable callus (9.63) (Figure 3b).

#### 2.2.6. Effect of the Interaction of the Medium, Phytagel, and Type of Callus

The analysis of the results showed that the number of embryos formed by the interaction between M2 medium (1.0 mg L^−1^ 2,4-D and 1.0 mg L^−1^ BAP), 5.0 g L^−1^ phytagel and the compact callus alone was significantly higher (127.47) than the combination of M1 medium (0.5 mg L^−1^ 2,4-D and 1.0 mg L^−1^ BAP), 2.3 g L^−1^ phytagel and friable callus, where the lowest number of embryos per gram of callus was produced (1.44) (Figure 4).

### 2.3. Histological Analysis of Embryogenic Calli

After 60 days of culture in the callus induction medium, the explants began to form two types of callus on the periphery and the midrib: friable or compact. The friable callus was beige, soft in texture, had a watery consistency, and was easily disintegrated (Figure 5a). This type of callus was made up of polyhedral parenchyma cells with abundant intercellular spaces and hardly evident nuclei (Figure 5c). The compact callus, for its part, was white in color, hard in texture, low in water content, and difficult to disintegrate (Figure 5b); this kind of callus was basically made up of elongated parenchyma cells with a certain degree of organization, small intercellular spaces, and hardly evident nuclei (Figure 5d).

Even though initially both types of callus were morphologically very different, after they were subcultured in the differentiation medium (8 weeks), both the friable and the compact calli lost turgor, turned dark brown (the result of the production of phenols), and began to form beige nodular structures (proembryogenic masses) (Figure 5e,f). These proembryogenic masses were made up of small cells with prominent nuclei, some degree of organization, and few intercellular spaces (Figure 5c,d). Four months after the calli were transferred to the differentiation medium, it was possible to observe the formation of the embryos from the proembryogenic masses (Figure 5g,h).

### 2.4. Development and Germination of Somatic Embryos

Embryos that were cultured in the globular state in the presence of 50 g L^−1^ sucrose, BAP, and IAA reached the cotyledon stage. Likewise, when these embryos were transferred to a mixture of substrates (perlite: vermiculite) in vitro, 55% of them germinated (Figure 6a). All regenerated seedlings survived when grown in the perlite:peat:volcanic rock mixture under greenhouse conditions (Figure 6b).

## 3. Discussion

The results obtained showed that the explants exposed to the different combinations of culture media (M1 and M2) and phytagel concentrations (2.3 and 5.0 g L^−1^) tested formed both friable (beige in color, soft in texture, watery, and easy to disintegrate) and compact callus (white in color, hard in texture, low in water content, and difficult to disintegrate). It is known that the primary calli characteristics may vary according to the species, the type of explant, and the culture conditions to which the explants are exposed. In this regard, López-Gómez et al. [10] reported the formation of two types of callus in the foliar explants of different genotypes of *Coffea* spp. The non-embryogenic callus was beige, and the embryogenic callus had a granular consistency and showed a dark yellow coloration. On the other hand, Padua et al. [12] found that the callus generated in the leaf explants of *C. arabica* cv. Catiguá cultured on a medium containing 2,4-D (0.5 mg L^−1^), indole butyric acid (IBA) (1.0 mg L^−1^), and 2iP (N6-2-isopentenyl adenine) (3.5 mg L^−1^), gave rise to two types of callus when transferred to another medium with BAP and 2,4-D. One of these types of callus was yellow and friable, and the other was transparent and watery. In *Coffea canephora*, a white and compact callus was observed on the explants established in 2,4-D, IBA, and 2iP. On transferring these calli to a medium with 2,4-D and BAP, they turned brown, and a yellow friable callus was generated after 10 weeks [19]. Conversely, the explants of the Caturra and Catuaí varieties of *C. arabica* formed compact and creamy primary callus on a medium containing 2,4-D, IBA, and 2iP; embryogenic callus (yellow and friable) originated on this callus when cultured with kinetin and 2,4-D [24]. In *Citrullus colocythis*, the formation of different types of primary callus was observed in leaf explants established in media containing different combinations of BAP, 2,4-D, naphthaleneacetic acid (NAA), and gibberellic acid (0.5–2.5 mg L^−1^). All types of callus formed embryos; however, those of the nodular, globular, or compact type showed a greater capacity to do so [31].

In the Colombia variety, two-thirds (68%) of the callus generated on the leaf explants was compact. The results indicate that the culture conditions tested in the present investigation (type and concentration of growth regulators and concentration of phytagel) were more conducive to the formation of compact callus than for friable callus. Although there are works in which the formation of different types of callus has been recognized in the explants cultivated in the media to induce somatic embryogenesis, in none of them was the proportion of each type of callus generated determined.

Furthermore, the results showed that friable and compact primary calli were formed in the same explant. This could be due to the fact that each type of callus could have been generated from different types of cells in the leaf explant. In this regard, Bartos et al. [21] observed that the primary calli of *C. arabica* cv. Catuaí, began to form from the vascular bundles (procambium) of the leaf explants. These authors also observed that the volume of palisade parenchyma cells close to the primary callus began to reduce until they acquired a flattened shape; however, the spongy parenchyma cells remained isodiametric. For their part, Menéndez-Yuffá and García [32] found that the callus was formed from both the perivascular tissue of the *C. arabica* leaf explants and the spongy parenchyma. The procambium is considered a meristematic tissue with high division capacity, and it is through it that growth regulators are transported [33]. Considering the above, it is possible to infer that procambium cells and parenchyma cells are not only different at an anatomical level but also at a physiological level, so callus generated from these types of cells could also show different characteristics.

Sugimoto et al. [34] demonstrated that callus is not only the result of somatic cell dedifferentiation but also of the presence of stem cells that seem to exist around vascular bundles in different types of organs. In the Colombia variety, callus formation was observed in the areas close to the vascular bundles of the leaf, but also in the periphery of the explant. This suggests that both stem cells and somatic differentiated cells exist in both friable and compact calli, some of which maintain their totipotency and capacity for the formation of somatic embryos (stem cells in the procambium), and others (parenchyma differentiated cells) acquire this capacity through the process of cellular dedifferentiation. Fehér [35] points out that stem cells can sense and respond to signals from the surrounding environment by producing molecules that determine cell fate and that these molecules can migrate to other regions of tissues. Other authors suggest that there could be components of the cell wall of embryogenic cells that migrate toward neighboring cells and confer embryogenic identity [36]. In this regard, arabinogalactan proteins (AGPs) make up a group of compounds that have been detected in the culture medium of different plant species. Van Hengel et al. [37] observed that adding AGPs to non-embryogenic cultures of *Daucus carota* promoted the formation of somatic embryos. In the same way, the number of somatic embryos increased when the *Triticum aestivum* cultures came into contact with AGPs [38]. This suggests that these compounds or others could migrate from stem cells present in both friable and compact varieties of the Colombia variety calli to change the fate of differentiated somatic cells, making them competent to form somatic embryos.

On the other hand, it was observed in this research that the number of embryos formed was higher as the phytagel concentration increased (5.0 g L^−1^). This response was observed when the effect of this gelling agent was analyzed independently or combined with the culture medium and the type of callus. This could be because phytagel is considered a matrix that makes water and nutrients less available to cells in the culture medium. The movement of water on the surface of the semi-solid culture medium is affected by the solute potential (Ψπ) and the matrix potential (Ψm); reducing either of the two potentials results in less movement of the water towards the surface, causing hydrostatic stress in cells [39]. Stress plays an important role in the induction of somatic embryogenesis since it promotes cell reprogramming and modifications in physiological and metabolic pathways [25,36,40,41]. Feher et al. [40] indicate that stress not only affects cell dedifferentiation but also promotes the formation of somatic embryos. Klimaszewska et al. [42] suggest that the higher concentration of the gelling agent reduces the availability of water, which stimulates a change in the development program of cells.

Conversely, even though no significant differences were found in the number of embryos that formed the calli of the Colombia variety cultured in the two culture media tested, the medium that contained equal parts of auxins and cytokinins (1 mg L^−1^ 2,4-D and 1 mg L^−1^ BAP) (M2) gave rise to a greater number of embryos than the medium that contained a higher ratio of cytokinins to auxins (0.5 mg L^−1^ 2,4-D and 1 mg L^−1^ BAP) (M1). In this regard, in the Ababuna hybrid of *C. arabica*, it was possible to obtain somatic embryos when the explants were cultivated in a medium with 1 mg L^−1^ of both BAP and 2,4-D, and then subcultured in the presence of high concentrations of BAP (4 mg L^−1^) [20]. In contrast, leaf explants of different cultivars of *C. arabica* (2000-692, 2000-1128) and *C. canephora* (97-20, 95-8, 00-28, 97-18) formed somatic embryos in the presence of 1.1 mg L^−1^ BAP and 0.5 mg L^−1^ 2,4-D [10].

Likewise, the fact that somatic embryos were obtained in two media tested in this research could be related to the presence of 2,4-D in both media. 2,4-D is one of the most commonly used growth regulators to induce the formation of embryogenic cells [43]. The presence of 2,4-D in the medium for induction of somatic embryogenesis is considered a key element in many species [36]. This compound, which is an auxin analog, is also considered a stress promoter; this characteristic contributes to its efficacy as an inducer of somatic embryogenesis [44]. The presence of 2,4-D in the culture medium leads to genetic reprogramming in somatic cells and the expression of hundreds of genes specifically required for the acquisition of embryogenic competence [45]. Studies utilizing AFLP molecular markers reveal that in *Coffea arabica*, the ability of cells to form somatic embryos is not due to changes at the DNA level but rather to methylation of this molecule (epigenetic changes). Methylation can change the genetic program of cells towards an embryogenic one through the differential expression of some genes (such as those related to stress) regulating their ability to form embryos [46]. It can be inferred that the presence of growth regulators such as 2,4-D, high concentrations of phytagel, and in vitro culture conditions *per se* contributed significantly to the acquisition of embryogenic competence in the somatic cells of *C. arabica* var. Colombia. Some authors consider that the interaction between auxins, cytokinins, and stress plays a central role in mediating the signal transduction cascade, allowing the reprogramming of gene expression, and acquiring the embryogenic competence of somatic cells. The reprogramming is followed by cell divisions that induce the formation of disorganized cells (callus) or polarized growth, which leads to the formation of somatic embryos [41].

Some authors point out that although auxins are essential for the induction stage, they can negatively affect embryonic differentiation. For this reason, after callus induction or multiplication, it is recommended to omit or decrease auxins in the culture medium to achieve differentiation and embryonic development [12,47]. This was confirmed in the present investigation since, when 2,4-D was removed from the medium, the embryos began to differentiate in the presence of BAP. Bai et al. [48] found that when the callus generated with 2,4-D is transferred to a medium free of it, endogenous auxin synthesis is induced at the periphery of the callus. The development of somatic embryos is related to the high concentration of exogenous and endogenous auxins. It is known that in the embryonic sac of the seeds, the egg cell forms in the area where auxins are in the greatest concentration (micropyle) [40].

On the other hand, the results showed that the type of callus had a significant effect on the formation of somatic embryos, with compact calli exhibiting the greatest ability to do so. A similar response was observed by Ramakrishna and Shasthree [31] in *Citrullus colocythis* and Bartos et al. [21] in *C. arabica* cv. Catuaí Vermelho, when they studied the embryogenic capacity of compact calli generated in different combinations of cytokinins and auxins. In *C. canephora* and *C. arabica* (cv. Caturra and Catuaí), it was also possible to regenerate somatic embryos from compact primary calli generated from leaf explants grown in 2,4-D, 2iP, and IBA [19,24].

Likewise, it was possible to observe that the combination of the M2 medium containing equal amounts of auxins and cytokinins (1 mg L^−1^ 2,4-D; 1 mg L^−1^ BAP) with 5.0 g L^−1^ phytagel and the compact callus generated the highest number of embryos per gram of callus. It should be noted that the number of somatic embryos that formed in all combinations of culture media, phytagel concentration, and type of callus tested was considerably higher when the compact callus was used. The above confirms that both types of calli can form embryos, but this capacity is greater in compact calli. In contrast, Toonen et al. (1994) [49] state that embryogenic capacity is not associated with a particular type of cell. At a morphological level, the compact callus seems to contain less water than the friable callus, a condition that could be related to its greater capacity to form embryos since, as mentioned above, the reduction in the amount of water in the cells can cause water stress, which induces physiological changes that could cause the reprogramming of cells to acquire embryogenic competence. The fact that the compact callus generated a greater number of somatic embryos in the Colombia variety could represent an advantage for the established protocol since two-thirds of the callus that formed the leaf explants was compact.

In the present research, it was possible to observe that only the cells that produced phenols (brown-colored callus) formed embryos. In this regard, Quiroz-Figueroa et al. [36] noted that the embryogenic cells of *C. arabica* showed brown coloration, while the pale-yellow cells did not form embryos. Moreover, in *Arbutus unedo,* only the calli that produced phenols formed embryos [50]. It is known that reactive oxygen species (ROS) are produced during the process of somatic embryogenesis, especially in the early stages. Phenols are compounds that can help cells neutralize the harmful effects of ROS produced in high concentrations [51,52]. However, ROS could have a positive effect on somatic embryogenesis, as evidenced in *Astralagus adsurgens* cultures, in which the endogenous increase in H_2_O_2_ induced an increase in embryo production [53]. In *Glycine max*, cells cultured with 2,4-D for the first two weeks showed overexpression of genes related to oxidative stress, programmed cell death, and cell division. Some authors propose that programmed cell death could be related to cell division and the correct formation of the development pattern of the embryo from proembryogenic masses [54,55]. In coffee, the production of phenolic compounds could be necessary for somatic embryogenesis and could even be used as a marker to recognize cells that can develop into somatic embryos [46].

The histological study of the cultures of the leaf explants of *C. arabica* var. Colombia allowed us to observe that the calli generated (friable and compact) were not only morphologically different but also anatomically different. Even though both types of callus consisted primarily of parenchyma cells, in friable calli the cells were polyhedral, with large spaces between them, as observed by Redway et al. [56] in friable calli of *Triticum aestivum* obtained from the culture of immature embryos. On the other hand, the parenchyma cells that made up the compact calli were elongated, flat, and had a certain degree of organization and compaction, which contrasts with the characteristics of the compact calli obtained in *Fagopyrum esculentum* in the presence of 2,4-D (2.0 mg L^−1^) and BAP (1.5 mg L^−1^). These calli had round or polyhedral cells, with large spaces between cells, inconspicuous nuclei, and some degree of compaction [57].

It may be noted that both friable and compact calli, once transferred to the differentiation medium, gave rise to clusters of cells with meristematic characteristics (proembryogenic masses). This type of cells were small, isodiametric, with a high division rate, few spaces between them, and very evident nuclei. From these, somatic embryos were differentiated, as observed by other authors in embryogenic cultures of *C. arabica* cvs. Caturra, Catuai and Catiguá [12,24,29,57].

Although various works mention the morphological distinctions between compact and friable calli, only a few reports have described the anatomical features of these calli [21,57]. These reports confirm that proembryogenic masses are produced from them, which further differentiate into somatic embryos.

Regarding somatic embryogenesis protocols in coffee, morphological, anatomical, or molecular markers are required to identify cells with the potential to form embryos from the initial stages of cultures. This is because it can take a long time (up to twelve months) to regenerate somatic embryos in this genus of plants. The brown coloration of calli can be considered an indicator to recognize those cells capable of giving rise to proembryogenic masses. However, this characteristic is evident after the primary calli are subcultured in the differentiation medium (90 to 120 days after starting the culture). In contrast, the morphological or anatomical characteristics of the primary calli of the Colombia variety of *C. arabica* could be used as markers that allow identifying cells with embryogenic potential from the callus induction stage (30 to 60 days after starting the culture).

The results obtained in the present investigation showed that the culture conditions tested promoted the formation of a primary callus that was friable and compact and made up of cells with different morphological and anatomical characteristics. Despite the differences observed between the different types of regenerated calli, it was possible to obtain somatic embryos from both. These findings make it possible to use both friable and compact primary calli produced during the embryogenesis induction stage. The above will save time, material, and labor, as well as increase the efficiency of the developed protocol by having a greater number of cells with embryogenic capacity.

## 4. Materials and Methods

### 4.1. Disinfestation

Young leaves (first and second pair) obtained from one-year-old plants grown in a greenhouse were immersed for 15 min in a 0.1% fungicide solution (Promyl^®^, Promotora Técnica Industrial, S.A de C.V, Morelos, México). Subsequently, the leaves were rinsed with sterilized distilled water and placed in a sodium hypochlorite solution (20% *v*/*v*) for 20 min, and then rinsed with sterilized distilled water.

### 4.2. Embryogenic Callus Induction

Explants (1 cm^2^ segments) were obtained from the disinfested leaves, which were placed in 90 × 15 mm Petri dishes with 30 mL of two induction media, which consisted of the basal salts of Murashige and Skoog (MS) [58], sorbitol (109 mg L^−1^), sucrose (30 g L^−1^), phytagel (2.3 and 5.0 g L^−1^), citric acid (100 mg L^−1^) and ascorbic acid (200 mg L^−1^). The M1 medium was supplemented with 0.5 mg L^−1^ 2,4-dichlorophenoxyacetic acid (2,4-D) and 1.0 mg L^−1^ 6-benzylamino purine (BAP); while the M2 medium contained 1.0 mg L^−1^ 2,4-D and 1.0 mg L^−1^ of BAP. The pH of the media was adjusted to 5.7 before being sterilized in an autoclave at 121 °C for 20 min. The cultures remained in a growth chamber at 26 ± 2 °C under darkness for two months. The experiment had a completely randomized design with two factors: culture media and phytagel. There was four treatments with 10 repetitions, one of which was a Petri dish with six explants.

### 4.3. Differentiation of Somatic Embryos

The calli formed on the explants cultivated in each of the induction media for eight weeks were classified as friable or compact. One gram of callus (friable or compact) was placed in 60 × 15 mm Petri dishes with 15 mL of differentiation medium; this medium contained the basal salts of Yasuda et al. [59] modified, 1.0 mg L^−1^ BAP, 30 g L^−1^ sucrose, 2.3 g or 5.0 g L^−1^ phytagel. The cultures were incubated at 26 ± 2 °C in darkness and after four months, the number of embryos per gram of callus was quantified. The experiment had a completely randomized factorial design with two factors: callus type (2) and phytagel concentration (2), giving rise to four treatments; each treatment consisted of 10 repetitions and one repetition consisted of a Petri dish with one gram of callus.

### 4.4. Statistic Analysis

In order to assess the treatments (phytagel-medio), a “Student’s t-test” was conducted for each of the four treatments tested, using a 5% significance level. Before analysis, the data were transformed using the arcsine function. To test the independent effect of factors such as phytagel concentration (2.3 and 5.0 g L^−1^), culture medium (M1 and M2), and the type of callus (friable and compact) on the number of embryos, the data were analyzed using the “F” test for variances of two samples. The difference between the means was compared using the “Student’s *t*-test” at a significance level of 5%. In order to examine how two and three factors interact, an analysis of variance (ANOVA) was conducted on the data. The treatment means were compared with a Tukey test with a 5% significance level. All analyses were performed using version 9.0 of the SAS statistical program.

### 4.5. Histology

Friable and compact embryogenic calli were fixed in a mixture of ethanol, water, formaldehyde, and acetic acid (52%, 33%, 10%, and 5%, respectively) (*v*/*v*); then they were dehydrated with different concentrations of ethanol (30, 40, 50, 70, 85, 100%), ethyl alcohol, and xylene (1:1) and pure xylene, and then embedded in paraffin. With a rotating microtome (American Optical^®^, Spencer 820 model, Vernon Hills, IL, USA), cuts (10 µm of thickness) were made, which were treated with xylene (100%) and 50, 70, 85, and 100% ethanol to eliminate the paraffin. The sections were stained with safranin O-fast green, infiltrated, and embedded in synthetic resin [60]. The samples were observed with an optical microscope (Carl Zeiss^®^, Tessovar Model, Oberkochen, Germany) and the images were captured with a Paxcam^®^ digital camera (Villa Park, IL, USA).

### 4.6. Development and Germination of Somatic Embryos

The globular stage embryos were cultured in 60 × 15 mm Petri dishes with culture medium (15 mL) containing the basal salts of Yasuda et al. [59] modified to 50 g L^−1^ sucrose, 0.25 mg L^−1^ BAP, 0.25 mg L^−1^ IAA, and 2.3 g L^−1^ phytagel. After three weeks, the embryos were transferred to flasks with seven mL of Yasuda et al. medium, with the same concentration of growth regulators and 30 g L^−1^ sucrose, without phytagel. The cultures were kept shaking at 100 rpm in a growth chamber at 26 + 2 °C and 16 h of fluorescent white light (60 µmol m^−2^ s^−1^). The embryos that germinated were cultured in 200 mL capacity flasks containing a mixture of vermiculite: perlite (3:1) with a particle size of 0.5 mm; the mixture of substrates was moistened with 30 mL of liquid MS medium at 50% of its concentration, 20 g L^−1^ sucrose, without growth regulators. The regenerated seedlings were transferred to a mixture of perlite:peat:volcanic rock (1:1:1) and were grown in the greenhouse.

## 5. Conclusions

The culture conditions (culture medium and phytagel concentration) established in the present investigation promoted the formation of different types of primary callus (friable and compact) from leaf explants of the Colombia variety of *C. arabica*. These calli were made up of cells with different morphological and anatomical characteristics. The friable calli were beige in color, soft in texture, watery, easy to disintegrate, and made up of polyhedral parenchyma cells with abundant intercellular spaces. The compact calli were white in color, hard in texture, had low water content, and disintegrated with difficulty; they were made of elongated parenchyma cells with large intercellular spaces and a certain degree of organization. The type of callus and the concentration of phytagel significantly affected the number of embryos that differentiated. Both friable and compact calli showed the capacity to form proembryogenic masses and somatic embryos, although the capacity of compact calli was greater. Combining 2,4-D and BAP (in equal proportion), 5.0 g L^−1^ phytagel, and compact callus allowed for the highest number of somatic embryos to be obtained. Considering that the cells of both friable and compact callus have embryogenic capacity, it is possible to use both types to obtain somatic embryos. The morphological and anatomical characteristics of the primary calli obtained can be used as markers that allow identifying cells with embryogenic potential, from the early stages of somatic embryogenesis.

## Figures and Tables

**Figure 1 plants-12-03570-f001:**
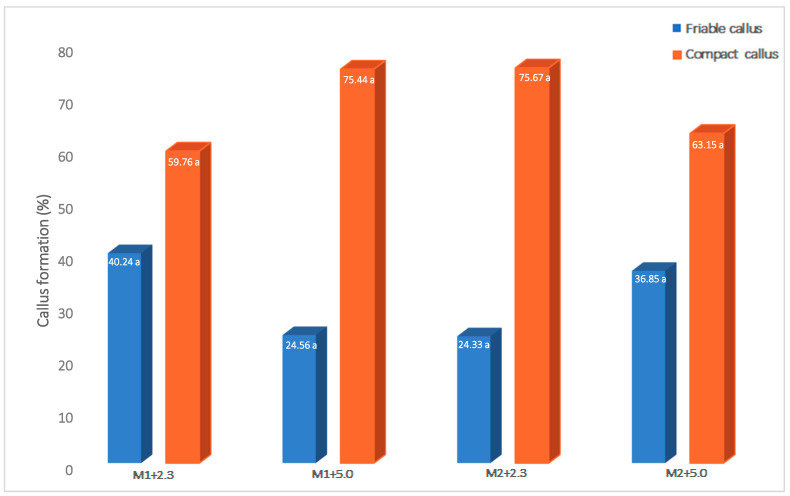
Effect of culture medium and phytagel on the calli characteristics generated from leaf explants of *Coffea arabica* var. Colombia. Means with the same letter within each combination (medium × phytagel) are not statistically different (Tukey α = 0.05). M1: MS salts + 0.5 mg L^−1^ 2,4-D + 1.0 mg L^−1^ BAP; M2: MS salts + 1.0 mg L^−1^ 2,4-D + 1.0 mg L^−1^ BAP; 2.3 or 5.0: phytagel concentration (g L^−1^).

**Figure 2 plants-12-03570-f002:**
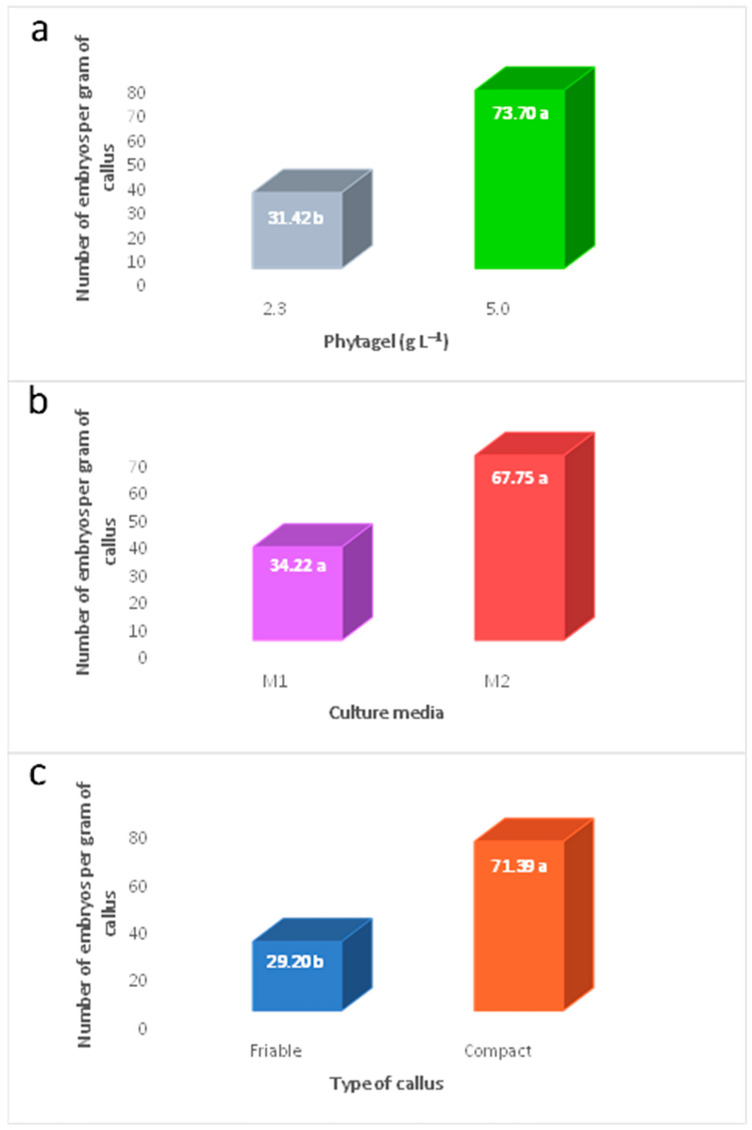
Number of embryos formed per gram of callus of *Coffea arabica* var. Colombia obtained by the independent effects of different factors. (**a**) Phytagel, (**b**) culture media, and (**c**) type of callus. Means with the same letter within the levels of each factor are not significantly different (Tukey, α = 0.05). M1: MS salts + 0.5 mg L^−1^ 2,4-D + 1.0 mg L^−1^ BAP; M2: MS salts + 1.0 mg L^−1^ 2,4-D + 1.0 mg L^−1^ BAP.

**Figure 3 plants-12-03570-f003:**
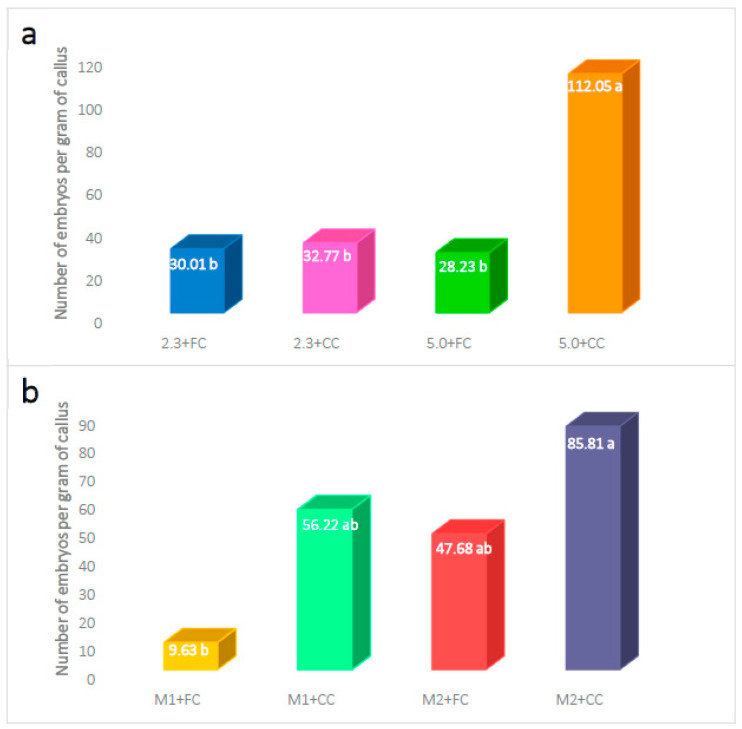
Effect of the interaction between two factors on the number of embryos formed per gram of callus of *Coffea arabica* var. Colombia. (**a**) Phytagel and callus type, and (**b**) culture medium and callus type, after six months of culture. Means with the same letter are not statistically different (Student t, α = 0.05); 2.3 or 5.0: phytagel concentration; M1: MS salts + 0.5 mg L^−1^ 2,4-D + 1.0 mg L^−1^ BAP; M2: MS salts + 1.0 mg L^−1^ 2,4-D + 1.0 mg L^−1^ BAP; FC: friable callus; CC: compact callus.

**Figure 4 plants-12-03570-f004:**
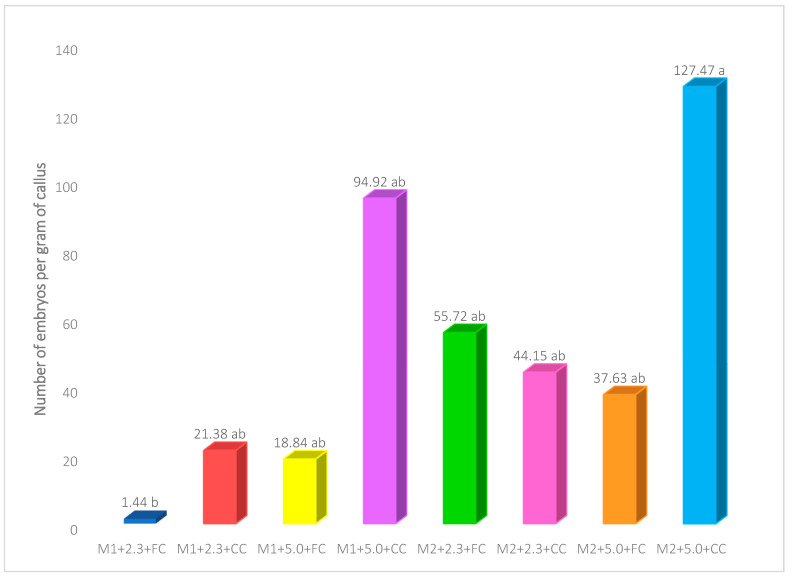
Effect of the interaction between culture medium, phytagel, and callus type on the number of embryos formed in *Coffea arabica* var. Colombia. Means with the same letter are not statistically different (Student t, α = 0.05). M1: MS salts + 0.5 mg L^−1^ 2,4-D + 1.0 mg L^−1^ BAP; M2: MS salts + 1.0 mg L^−1^ 2,4-D + 1.0 mg L^−1^ BAP; 2.3 or 5.0: phytagel concentration (g L^−1^); FC: friable callus; CC: compact callus.

**Figure 5 plants-12-03570-f005:**
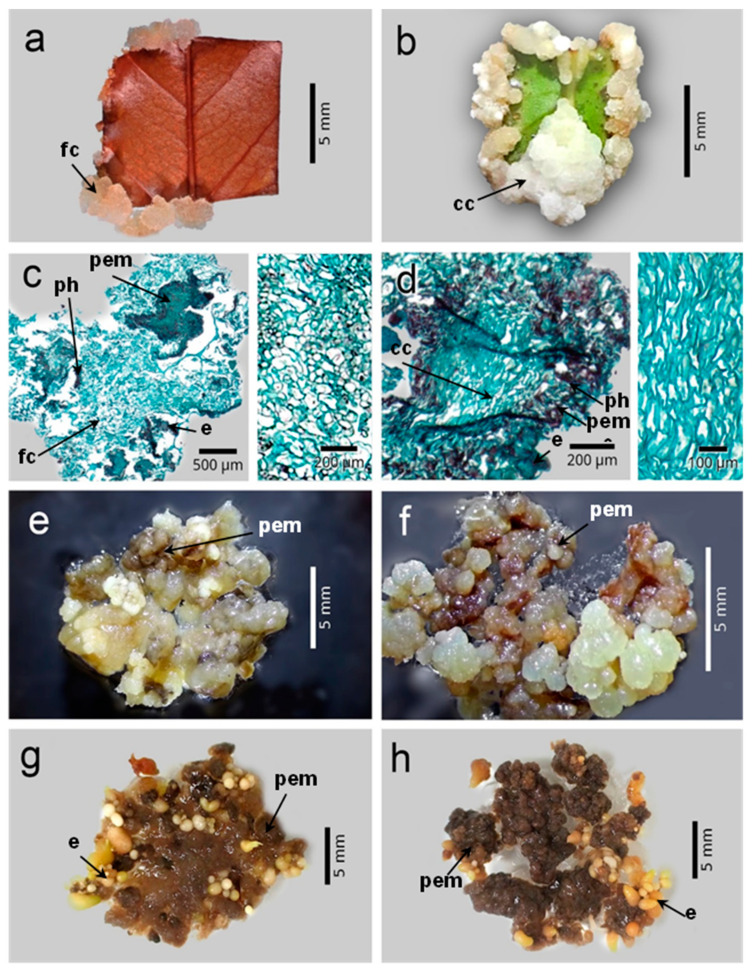
Types of embryogenic callus in *Coffea arabica* L. var. Colombia. (**a**) Friable callus at six weeks of culture. (**b**) Compact callus eight weeks after starting the induction stage. (**c**) Histology of friable callus (right, close-up). (**d**) Histology of compact callus (right, close-up). (**e**) Formation of proembryogenic masses on the friable callus. (**f**) Compact callus with proembryogenic masses. (**g**) Formation of embryos at six months of culture from friable callus. (**h**) Embryos formed on compact embryogenic callus after six months of culture. fc: friable callus; cc: compact callus; pem: proembryogenic masses; ph: phenols; e: embryos.

**Figure 6 plants-12-03570-f006:**
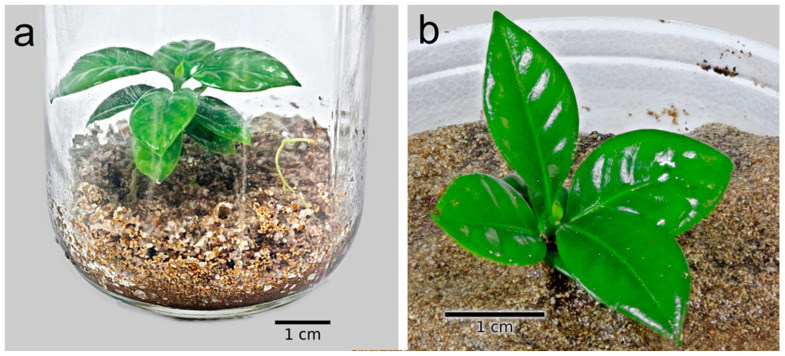
Plants of *Coffea arabica* L. var. Colombia regenerated by indirect somatic embryogenesis. (**a**) Seedlings in a mixture of perlite:vermiculite under in vitro conditions. (**b**) Plants growing in a greenhouse in a mix of perlite:peat:volcanic rock.

## Data Availability

All data generated in this study are included in the tables and figures.

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
