# Peer review of "Callus Type, Growth Regulators, and Phytagel on Indirect Somatic Embryogenesis of Coffee (Coffea arabica L. var. Colombia)"

_plants, 2023, doi:10.3390/plants12203570_

Round 1

Reviewer 1 Report

Comments for Author,

I have read your paper carefully. This paper is interesting, however, I can not suggest this paper for the publication because of lack of novelty, poorly quality of discussion. However, I have some minor comments 

Please add more data to improve abstract section.

Introduction is so general. Please add more properly refs. 

Material and methods are clearly explained. If possible, please extend statistical part. 

Please extend introduction. 

Discussion is well documented. Please more discuss obtained results carefully.

Conclusion remark is missed. Please add your future recommandation.

Best Regards

Reviewer 2 Report

The manuscript presents an interesting study on the influence of callus type, growth regulators, and phytagel concentration on the somatic embryogenesis of Coffea arabica var. Colombia. However, there are several areas where clarity and organization can be improved.

The introduction should provide more context about the significance of somatic embryogenesis in coffee production and why it's important to study the factors mentioned in the research. Additionally, the introduction could benefit from a clear statement of the research objectives.
L62: please support the sentence with appropriate citations (DOI: 10.1007/s00253-021-11375-y; 10.1186/s12896-023-00796-4)

Results are presented somewhat ambiguously. It would be helpful to include figures to illustrate the data and trends clearly. Additionally, the discussion of the results is limited and lacks an in-depth analysis of the implications of the findings.

The manuscript mentions two types of callus, friable and compact, but does not provide a clear definition or characterization of these types. It's important to describe these types in detail and provide criteria for distinguishing between them. Additionally, the terminology used (e.g., "beige" and "easily breakable" for friable callus) should be standardized and made more precise.

The conclusion section is relatively brief and does not fully explore the implications of the findings. It should delve into the practical applications of the research results in coffee production and discuss how the findings contribute to the existing knowledge in the field of somatic embryogenesis.

Overall, this manuscript has the potential to contribute significantly to the understanding of somatic embryogenesis in coffee, but it requires substantial revisions to enhance clarity, rigor, and scientific depth.

Reviewer 3 Report

Manuscript Callus Type, Growth Regulators, and Phytagel on Indirect Somatic Embryogenesis of Coffee (Coffea arabica L. var. Colombia) by Consuelo Margarita Avila-Victor , Enrique de Jesús Arjona-Suárez , Leobardo Iracheta-Donjuan , Jorge Manuel Valdez-Carrasco , Fernando Carlos Gómez-Merino, Alejandrina Robledo-Paz reviews somatic embryogenesis protocols potentially useful for the production of transgenic or transplastomic plants.

The manuscript is formatted according to the rules, well illustrated and contains the necessary sections.

The article is presented with high-quality illustrations and accompanied by statistical data.

The article may be accepted.

Small notes:

Keywords should include in vitro culture, not just culture.

On pp. 320-321, replace the underscore with a minus sign.

In line 383, replace the test student's t with Student if you meant the last name?

Round 2

Reviewer 2 Report

All the comments have been addressed. I think that the current version of the manuscript can be published in the journal.